# Factors associated with abnormal left ventricular ejection fraction (decreased or increased) in patients with sepsis in the intensive care unit

**Dong Geum Shin[1], Min-Kyung Kang[1] *, Yu Bin Seo[2], Jaehuk Choi[3], Seon Yong Choi[1], Seonghoon Choi[1], Jung Rae Cho[1], Namho Lee[1]**

**1** Division of Cardiology, Kangnam Sacred Heart Hospital, Hallym University Medical Center, Seoul, South Korea, **2** Division of Infection, Kangnam Sacred Heart Hospital, Hallym University Medical Center, Seoul, South Korea, **3** Division of Cardiology, Dongtan Sacred Heart Hospital, Hallym University Medical Center, Seoul, South Korea

* homes78@naver.com

## Abstract

### Background

Sepsis-induced cardiomyopathy (SIC) is known to show cardiac dysfunction in patients with sepsis. Both a decrease or an increase in ejection fraction (EF), an indicator of cardiac function, can occur. The purpose of this study was to identify factors associated with abnormal left ventricular (LV) function measured by EF in patients with sepsis in the intensive care unit (ICU).

### Methods

This was a retrospective study performed from November 2016 to December 2018. Three-hundred and sixty-six patients (mean age, 73 ± 13 years; 191 [52%] men) admitted to the ICU with sepsis were included. Patients were classified into three categories according to LV EF (group 1 –[EF<50%, n = 36], group 2 –[50≤EF<70%, n = 252], and group 3 – [EF≥70%, n = 78]). Echocardiographic assessment was performed within 48 hours of diagnosis of sepsis. We analyzed clinical factors including mortality, echocardiographic findings, and laboratory parameters.

### Results

Decreased LV EF occurred in 36 (10%) patients and hyper-dynamic EF developed in 78 (21%) patients. Of 366 patients, 103 (28%) patients died. Baseline characteristics were similar in the three groups, except female sex an indicator of abnormal EF. Mortality rates were also similar in the three groups; however, mortality rates were significantly higher in patients with abnormal EF (decreased or increased vs. normal). Echocardiographic parameters were significantly different in the three groups, in terms of LV systolic parameters and chamber size. Small left atrium (LA) and small LV were significantly associated with abnormal EF

**Data Availability Statement:** All relevant data are within the paper and its Supporting Information files.

**Funding:** This research received no specific grant from any funding agency.

**Competing interests:** The authors have declared that no competing interests exist.

(especially in patients with increased EF). High brain natriuretic peptide was associated with decreased EF. Among these factors, female sex and small LA were significantly associated with abnormal EF in the multiple regression analysis.

## Conclusion

Our findings highlight that female sex and small cardiac size are associated with abnormal EF, and therefore, death. Therefore, female patients and patients with small LA should be monitored closely when they present with sepsis.

## Introduction

Sepsis is a lethal syndrome induced by infection which is associated with a high mortality rate, and in fact, it is the main cause of death in non-cardiac intensive care units (ICU) [1]. The pathophysiology of sepsis includes inflammation, immune dysfunction, and coagulation disorders. The septic shock in the early onset of sepsis is a main cause of death for septic patients [2]. Septic shock is known to be caused mainly by immunosuppression that occurs in the late stage; however, cytokine storm and cardiac dysfunction are also main causes of septic shock in the early onset of sepsis [3]. The heart plays a key role in the pathophysiology of septic shock, and therefore, it is important to evaluate myocardial function and continue hemodynamic monitoring in patients with sepsis. Conventionally, the definition of sepsis-induced cardiomyopathy (SIC) is a global, reversible, systolic and diastolic dysfunction of the left ventricle (LV) or right ventricle (RV), which is induced by myocardial depressants released from pathogens and the host, and global ischemia after peripheral vasodilation, and arterial and capillary shunting in septic distributive shock [3]. A retrospective cohort study reported that SIC developed in 13.8% of patients with sepsis and septic shock [4], so SIC could be used as an outcome predictor in septic patients [5]. The mainly therapy for SIC focuses on achieving hemodynamic stabilization using fluid therapy [6], inotropic drugs [7], or immunomodulation [8–10]. Inotropic therapy is suggested for patients with low cardiac output after proper fluid therapy. Norepinephrine is the first choice recommended by the guideline, while the use of dobutamine and dopamine is recommended only for selected patients due to the associated adverse events [7]. Therefore, patients with SIC need a different approach to patients with conventional cardiogenic shock or heart failure.

Contrary to SIC, hyperdynamic LV ejection fraction (EF), which is defined as a LVEF > 70%, is frequently observed on transthoracic echocardiography (TTE) in the ICU [11]. Patients with sepsis commonly have low systemic vascular resistance and increased circulating catecholamines, and this causes increased contractility [12]. In terms of prognosis, a recent meta-analysis on sepsis-induced LV dysfunction showed there is no significant correlation between reduced EF and mortality [13]. Rather, hyperdynamic LV is a possible predictor of mortality in patients with sepsis combined with a high APACHE II score [14].

Therefore, the purpose of this study was to identify the incidence of abnormal EF (reduced or increased) in patients admitted to the ICU with sepsis, and to investigate the differences in in-hospital mortality rates and the related clinical factors.

## Methods

### Study design and participants

This study was a retrospective study. In this study, 366 patients (191 [52%] male, average age: 73 ± 13 years) who attended the Kangnam Sacred heart Hospital, Hallym University from the

November 2016 to December 2018. Among the patients hospitalized with sepsis, we included patients were admitted with in the ICU and performed echocardiography, and refer to division of cardiology for evaluation of cardiac function. Patients with sepsis were diagnosed according to the definition of Sepsis-3 [15]. Patients with documented acute coronary syndrome (ACS), preexisting ischemic cardiomyopathy (ICMP), hypertrophic CMP (HCMP), restrictive CMP (RCMP), or other CMP with preexisting LV dysfunction, suspected stress induced CMP, previous cardiac surgery, significant valvular dysfunction, pulmonary thromboembolism, infective endocarditis, or chronic kidney diseases with dialysis were excluded from this study (Fig 1).

All procedures performed in studies involving human participants were in accordance with the ethical standards of the institutional and/or national research committee and with the 1964 Helsinki declaration and its later amendments or comparable ethical standards. Informed consent was not obtained from all individual participants included in the study, because this study was a retrospective study. There was no information the authors had access to potentially identifying patient information. This study was approved by the Institutional Review Board of Hallym University Kangnam Sacred Hospital (IRB no. 2019-02-005).

## Classification of the participants

I—Patients were classified into the three categories (decreased, normal, or hyper-dynamic EF) according to the LV EF (Fig 2).

1. Group 1 –decreased LV EF (EF < 50%) = 36 (10%),

2. Group 2 –normal LV EF (50≤ EF <70%) = 252 (79%),

3. Group 3 –hyper-dynamic LV EF (EF ≥70%) = 78 (21%)

II—We classified patients with EF less than 50% or 70% or more as abnormal EF vs. normal EF (EF 50≤ EF <70%).

1. Abnormal EF (AEF, group 1 and 3) = 114 (21%),

2. Normal EF (NEF, group 2) = 252 (79%)

Then we compared mortality rate (in-hospital mortality) and other clinical, laboratory and echocardiographic parameters between groups. Echocardiographic assessment was done within 48 hours of diagnosis of sepsis. Immediately after the visit, the blood test was performed as soon as possible and the follow-up test was carried out if necessary. The first CRP was CRPi (initial CRP), the highest was CRPp (peak CRP), and the last CRP before discharge or death was CRPf (final CRP). If possible, cardiac troponin was also tested up to three times or more if necessary. $TnI_1$ (cardiac troponin 1) was the first test, $TnI_2$ was the second test, and $TnI_3$ was the third or significant last one.

## Echocardiography

TTE was performed using standard techniques with a 2.5-MHz transducer. TTE was performed by well-trained sonographer (over 6 months) and it was confirmed by a cardiologist almost in real time. The standard 2-D and Doppler echocardiography was performed using a commercially available echocardiographic machine (Vivid 7R GE Medical System, Horten, Norway) with the same setup interfaced with a 2.5-MHz phased-array probe. All measurements were performed according to the guideline [16]. With the study participant in the partial left decubitus position and breathing normally, the observer obtained images from the

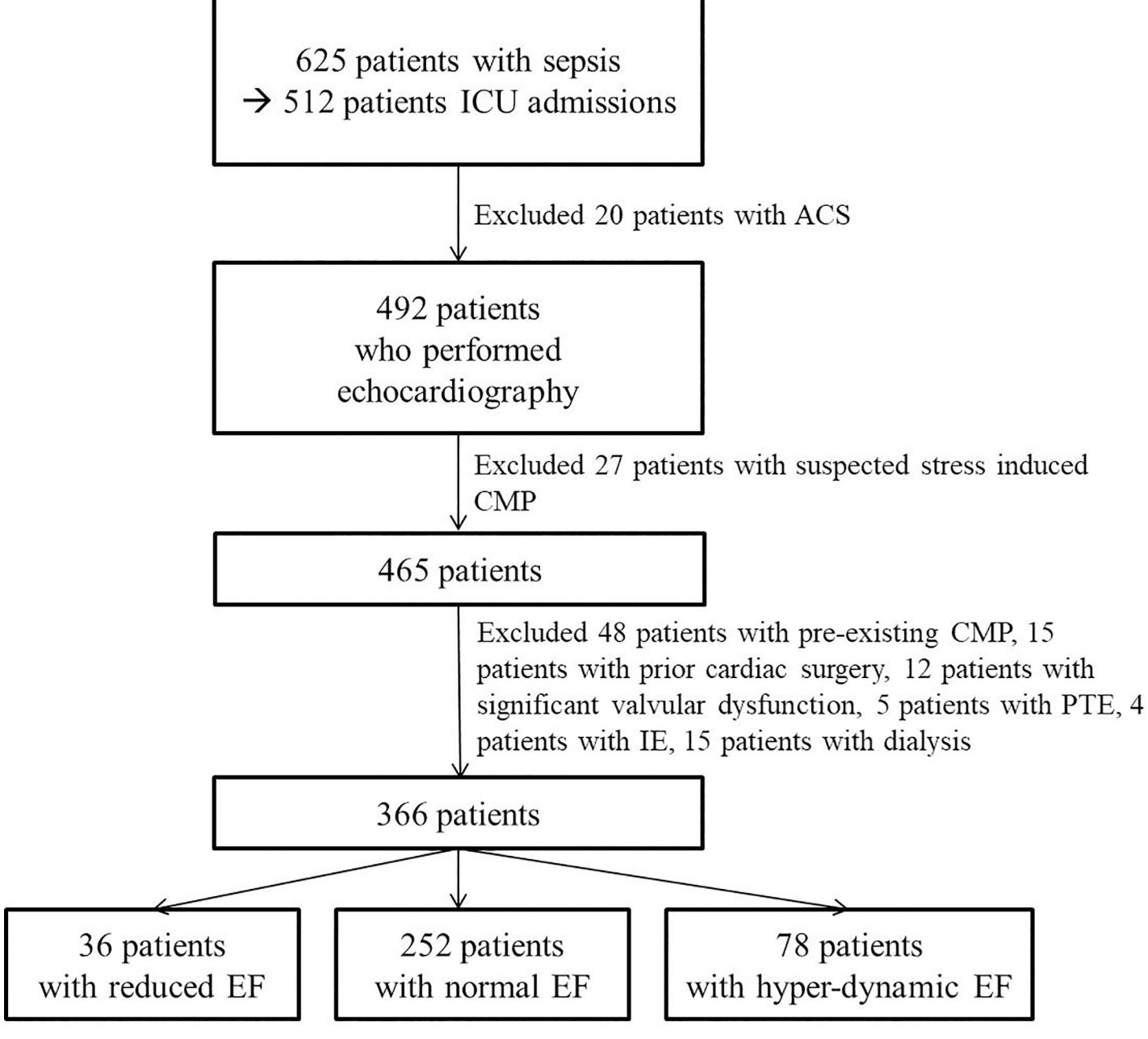

**Fig 1. Flowchart showing selection of patients in this study.** ICU: intensive care unit, ACS: acute coronary syndrome, CMP: cardiomyopathy, PTE: pulmonary thromboembolism, IE: infective endocarditis (some patients were overlapped).

parasternal long and short axes and from the apical four chamber and two-chamber and long-axis views. Depth setting was optimized to display the LV on the screen as large as possible and the same field depth was kept for both four and two-chamber apical views. Sector width was reduced to increase spatial and temporal resolution. LV end-diastolic dimensions (LV EDD), end-diastolic interventricular septal thickness, and end-diastolic LV posterior wall thickness were measured at end-diastole according to the standards established by the American Society of Echocardiography. LV EF was determined by the biplane Simpson's method. Maximal left atrial (LA) volume was calculated using the Simpson method and indexed to the body surface

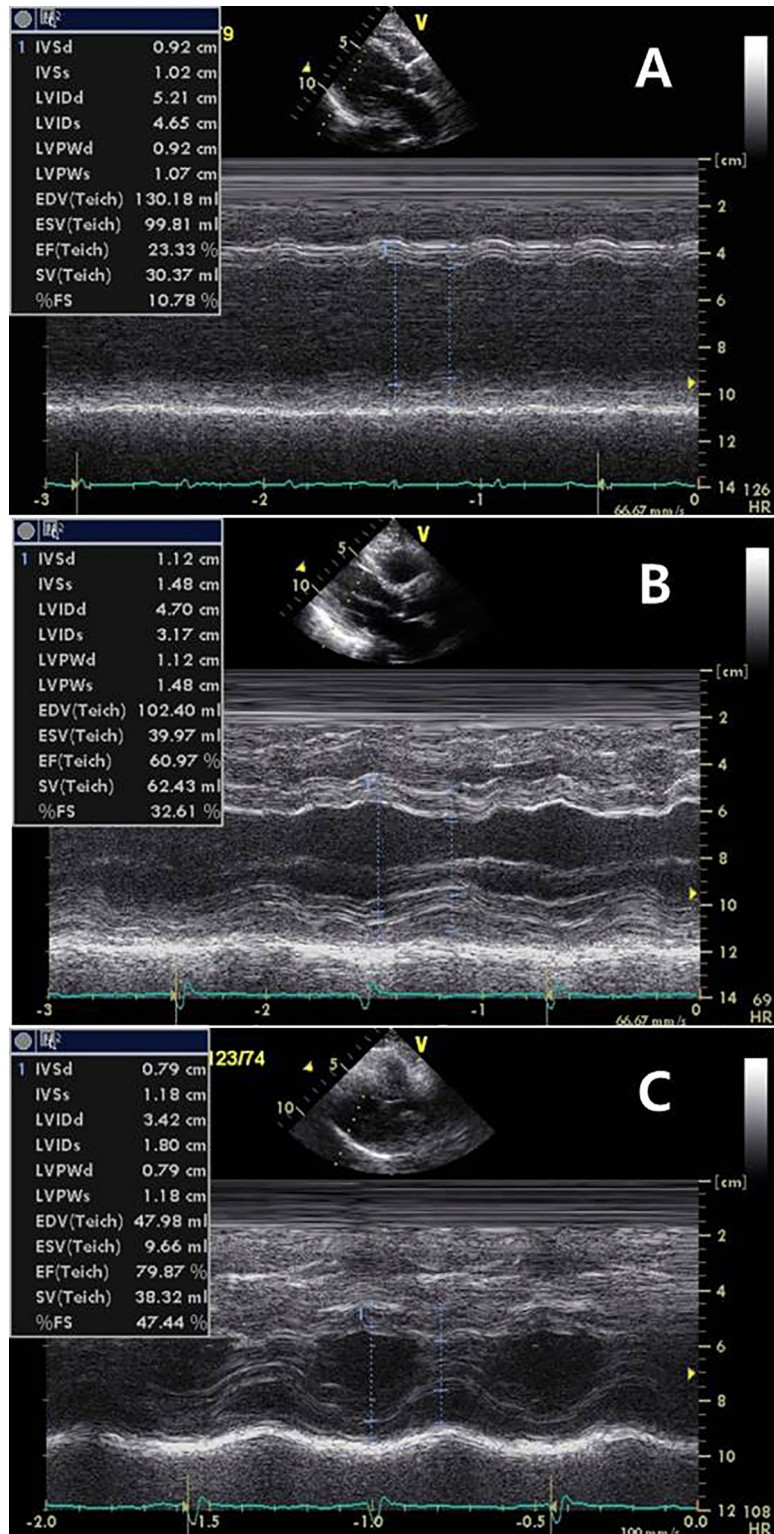

**Fig 2. M-mode echocardiographic findings of patients with sepsis. A:** a patient with reduced EF (EF: 25%), **B:** a patient with normal EF (EF: 66%), **C:** a patient with increased EF (EF: 79%). EF: ejection fraction (measured by biplane method).

area. LV mass was calculated using the Devereux formula = $1.04[(LVEDD + IVSTd + PWTd)^3 - (LVEDD)^3] - 13.6$. Thereafter, the LV mass index (LVMI) was calculated and indexed to body surface area. DWS was calculated as [(PWTs)—(PWTd)/(PWTs)] using M-mode echocardiography.

Mitral flow velocities were recorded in the apical four-chamber view. Mitral inflow measurements included the peak early (E) and peak late (A) flow velocities and the E/A ratio. The tissue Doppler of the mitral annulus movement was also obtained from the apical four-chamber view. A 1.5-mm sample volume was placed sequentially at the septal annular sites. The analysis was performed for early diastolic (E'), late diastolic (A') and systolic (S') peak tissue velocities. As a noninvasive parameter for LV stiffness, the LV filling index (E/E') was calculated by the ratio of transmitral flow velocity to annular velocity. Adequate mitral and tissue Doppler image (TDI) signals were recorded in all patients.

Longitudinal global strain (GS) of LV was obtained from apical 4, 3, 2–chamber views by speckle-tracking 2D-strain imaging [17].

Myocardial performance index (MPI, or tei index) was calculated as the sum of isovolumic contraction time and isovolumic relaxation time divided by the ejection time [18].

## Statistical analysis

All continuous data are expressed as mean ± SD, and all categorical data are presented as percentage or absolute numbers. Continuous variables were analyzed using one way ANOVA in three independent groups, and Student's t-test in two independent groups and dichotomous variables were analyzed using the chi square test. Non-normally distributed variables were analyzed using the Kruskal-Wallis test or Mann-Whitney U test. Cox regression analysis was performed to evaluate significant variables associated with abnormal EF and hyper-dynamic EF. P-value of less than 0.05 was considered statistically significant. All tests were performed using SPSS, version 18.0 (SPSS, Inc., Chicago, IL, USA). In addition, we performed Bonferroni correction for the clinically and statistically significant parameters (female gender, length of hospitalization, and left atrial volume index) between the three groups.

## Results

### Clinical parameters of the study population (Table 1)

There were no significant differences in age, blood pressure, underlying diseases, general condition, or site of infection among the three groups. The percentage of females was higher among patients in groups 1 and 3 compared to group 2. There was no statistically significant difference in in-hospital mortality rates among the three groups. The length of hospitalization was significantly shorter in patients with hyperdynamic EF.

### Laboratory findings and other parameters (Table 2)

There were no significant intergroup differences for chemistry, cardiac biomarkers, and inflammatory markers, and so on. However, brain natriuretic peptide (BNP) and initial troponin were the highest in patients with reduced EF. Non-normally distributed variables, including BNP and troponin, were analyzed using the Kruskal-Wallis test. The results showed that only BNP was significantly higher in patients with reduced EF.

### Echocardiographic parameters of the study population (Table 3)

Cardiac chamber size was the smallest in patients with hyperdynamic LV, and LV size was the largest in patients with reduced EF. However, LAVI was less in patients with reduced EF than

**Table 1. Clinical parameters of the study population.**

| | Group 1 (n = 36) | Group2 (n = 252) | Group3 (n = 78) | p |
|---|---|---|---|---|
| Age (years) | 75.4 ± 12.5 | 72.3 ± 13.5 | 72.8 ± 13.0 | 0.388 |
| Female gender* | 21 (58%) | 109 (43%) | 45 (58%) | 0.034 |
| SBP (mmHg) | 124 ± 24 | 123 ± 21 | 121 ± 21 | 0.821 |
| DBP | 70 ± 15 | 69 ± 14 | 67 ± 14 | 0.538 |
| Heart rate (bpm) | 98 ± 25 | 91 ± 20 | 93 ± 22 | 0.055 |
| Mortality | 13 (36%) | 63 (25%) | 27 (35%) | 0.137 |
| Etiology of infection | | | | 0.364 |
| Pneumonia | 19 (53%) | 95 (38%) | 36 (46%) | |
| GI | 2 (6%) | 38 (15%) | 10 (13%) | |
| UTI or APN | 8 (22%) | 68 (27%) | 21 (27%) | |
| Musculoskeletal | 1 (4%) | 19 (8%) | 3 (4%) | |
| Abscess | 5 (13%) | 20 (8%) | 3 (4%) | |
| Et al. | 1 (3%) | 12 (5%) | 5 (6%) | |
| Bed-ridden state | 7 (19%) | 73 (29%) | 22 (28%) | 0.490 |
| Nursing facility | 13 (36%) | 69 (28%) | 20 (26%) | 0.491 |
| Hypertension | 14 (39%) | 105 (42%) | 40 (51%) | 0.276 |
| Diabetes mellitus | 11 (31%) | 71 (28%) | 23 (30%) | 0.943 |
| Atrial fibrillation | 7 (19%) | 36 (14%) | 10 (13%) | 0.638 |
| Cerebrovascular ds | 14 (39%) | 96 (38%) | 28 (36%) | 0.929 |
| Cardiovascular ds | 7 (19%) | 30 (12%) | 5 (6%) | 0.118 |
| Pulmonary ds | 2 (6%) | 31 (12%) | 13 (17%) | 0.246 |
| Liver ds | 1 (3%) | 25 (10%) | 7 (9%) | 0.372 |
| Musculoskeletal ds | 9 (25%) | 52 (21%) | 13 (17%) | 0.561 |
| Recent surgery | 3 (8%) | 16 (6%) | 2 (3%) | 0.353 |
| Malignancy | 5 (14%) | 46 (18%) | 19 (24%) | 0.342 |
| SAPS3 | 36.3 ± 5.4 | 35.3 ± 7.6 | 34.3 ± 7.3 | 0.369 |
| Length of ICU | 8.8 ± 11.1 | 12.9 ± 19.6 | 8.3 ± 8.9 | 0.071 |
| Length of hospitalization* | 23.5 ± 22.7 | 25.5 ± 22.2 | 17.9 ± 14.0 | 0.019 |
| Initial CVP | 5.8 ± 2.9 | 6.0 ± 3.6 | 5.1 ± 3.2 | 0.276 |

Data are mean ± standard deviation (SD) or n (%). SBP: systolic blood pressure, DBP: diastolic BP, GI: gastrointestinal, UTI: urinary track infection, APN: acute pyelonephritis, ds: diseases, ICU: intensive care unit, CVP: central venous pressure.

*Bonferroni correction was done; p value for female gender- 1 vs. 2: 0.107, 1 vs. 3: 1.000, 2 vs. 3: 0.028, p value for length of hospitalization—1 vs. 2: 0.615, 1 vs. 3: 0.106, 2 vs. 3: 0.005.

in patients with normal EF. GS and S' velocity, which were related to the LV systolic function, were similar to EF in the three groups. There was no significant difference in terms of diastolic function between the three groups. MPI, an indicator of LV systolic and diastolic function, was also lowest in patients with hyperdynamic EF and highest in patients with reduced EF.

## Comparison of results of patients with normal vs. abnormal EF (NEF vs. AEF)

Table 4 shows clinical, laboratory, and echocardiographic parameters according to the presence of hypo (EF < 50%)—or hyperdynamic EF (EF ≥70%). The proportion of females was higher in patients with AEF, and the mortality rate was also higher (Fig 3). Heart rate was slightly but significantly faster in patients with AEF. Days of hospitalization and ICU were significantly shorter in patients with AEF. No significant differences were seen in laboratory

**Table 2. Laboratory parameters.**

| | Group 1 (n = 36) | Group2 (n = 252) | Group3 (n = 78) | p |
|---|---|---|---|---|
| Serum cr (mg/dL) | 1.4 ± 1.4 | 1.5 ± 1.2 | 1.5 ± 1.2 | 0.928 |
| BNP | 944 ± 1315 | 374 ± 981 | 344 ± 1543 | 0.001 |
| CK-MB | 14.8 ± 26.7 | 6.9 ± 15.6 | 8.3 ± 22.4 | 0.062 |
| $TnI_1$ | 1.17 ± 2.87 | 0.21 ± 0.81 | 0.18 ± 0.62 | <0.001 |
| $TnI_2$ | 4.59 ± 11.46 | 1.33 ± 4.08 | 0.88 ± 2.08 | 0.019 |
| $TnI_3$ | 1.00 ± 1.80 | 0.92 ± 5.81 | 0.66 ± 1.28 | 0.968 |
| CRPi | 128 ± 97 | 152 ± 108 | 148 ± 114 | 0.472 |
| CRPp | 192.2 ± 82.7 | 221.5 ± 144.7 | 224.0 ± 111.5 | 0.198 |
| CRPf | 61 ± 75 | 76 ± 270 | 73 ± 232 | 0.942 |
| D-dimer | 7.3 ± 8.5 | 7.2 ± 15.4 | 5.4 ± 4.8 | 0.798 |
| Procalcitonin | 18.3 ± 27.3 | 28.3 ± 114.6 | 13.5 ± 40.2 | 0.505 |
| BNP* | 179 (2.40–13210.00) | | | 0.001 |
| CK-MB* | 2.21 (0.02–154.21) | | | 0.094 |
| $TnI_1$* | 0.320 (0.006–13.185) | | | 0.417 |
| $TnI_2$* | 0.146 (0.006–47.633) | | | 0.368 |
| TnI3* | 0.077 (0.006–49.996) | | | 0.294 |

Data are expressed as mean ± SD. cr: creatinine, BNP: B-type natriuretic peptide, CK-MB: creatine kinase-muscle/brain, TnI: cardiac-specific troponin I (1,2,3 for [1st], [2nd], [3rd]), CRP: C-reactive protin (i-initial, p-peak, f-final * Analyzed by Kruskal-Wallis test and data are expressed as median (min~max)

**Table 3. Echocardiographic parameters of the study population.**

| | Group 1 (n = 36) | Group2 (n = 252) | Group3 (n = 78) | p |
|---|---|---|---|---|
| LAVI (ml/m$^2$) * | 22.9 ± 13.8 | 23.0 ± 13.7 | 17.3 ± 10.0 | 0.006 |
| LVMI (g/m$^2$) | 103.8 ± 28.7 | 96.7 ± 27.5 | 87.8 ± 21.2 | 0.005 |
| LVEDD (mm) | 48.8 ± 6.4 | 45.8 ± 6.3 | 43.1 ± 6.7 | <0.001 |
| ESD | 38.8 ± 6.7 | 30.5 ± 4.5 | 24.8 ± 3.7 | <0.001 |
| LV EF (%) | 40.6 ± 6.4 | 62.0 ± 4.5 | 73.1 ± 2.8 | <0.001 |
| GS (%) | -12.6 ± 4.4 | -17.3 ± 2.8 | -17.1 ± 4.6 | 0.004 |
| MPI | 0.52 ± 0.22 | 0.43 ± 0.20 | 0.40 ± 0.200 | 0.022 |
| E (cm/s) | 69.7 ± 28.7 | 70.4 ± 22.6 | 70.8 ± 21.2 | 0.978 |
| A (cm/s) | 77.9 ± 28.1 | 83.3 ± 23.3 | 87.6 ± 24.4 | 0.213 |
| E/A ratio | 0.98 ± 0.65 | 0.86 ± 0.35 | 0.85 ± 0.33 | 0.355 |
| DT (ms) | 167.1 ± 44.2 | 187.6 ± 46.2 | 185.9 ± 46.3 | 0.088 |
| E/E' | 13.5 ± 4.5 | 12.3 ± 5.2 | 11.6 ± 5.8 | 0.281 |
| S' (cm/s) | 5.9 ± 1.6 | 7.9 ± 2.4 | 8.6 ± 2.5 | <0.001 |
| RVSP | 35.6 ± 10.4 | 33.4 ± 11.2 | 34.8 ± 10.5 | 0.409 |
| Diastolic gr. | | | | 0.072 |
| normal | 2 (9%) | 19 (10%) | 6 (10%) | |
| Grade 1 | 17 (74%) | 153 (81%) | 51 (82%) | |
| Grade 2 | 3 (13%) | 18 (10%) | 5 (8%) | |
| Grade 3 | 1 (4%) | 0 (0%) | 0 (0%) | |

Data are represented as mean ± SD or n (%). LAVI: left atrial volume index, LVMI: left ventricular mass index; LV EDD and ESD: LV end-diastolic and systolic dimension, EF: ejection fraction, GS: global strain, MPI: myocardial performance index, DT: deceleration time, RVSP; right ventricular systolic pressure, gr: grade. *Bonferroni correction was done; p value for LAVI—1 vs. 2: 0.973, 1 vs. 3: 0.021, 2 vs. 3: 0.002.

**Table 4. Comparison of results of patients with normal vs. abnormal EF.**

| | AEF (n = 114) | NEF (n = 252) | p |
|---|---|---|---|
| Age (years) | 73.9 ± 11.6 | 72.3 ± 13.5 | 0.278 |
| Female gender | 66 (58%) | 109 (43%) | 0.013 |
| Mortality | 40 (35%) | 63 (25%) | 0.050 |
| SBP (mmHg) | 122 ± 22 | 123 ± 21 | 0.751 |
| DBP | 68 ± 15 | 69 ± 14 | 0.697 |
| Heart rate (bpm) | 97 ± 25 | 91 ± 19 | 0.030 |
| Body mass index (kg/m$^2$) | 22.0 ± 4.5 | 22.7 ± 4.3 | 0.158 |
| Site of infection | | | 0.353 |
| Pneumonia | 55 (48%) | 95 (38%) | *0.066 |
| Hypertension | 54 (47%) | 105 (42%) | 0.362 |
| Diabetes | 34 (30%) | 71 (28%) | 0.803 |
| Cerebrovascular ds | 42 (37%) | 96 (38%) | 0.907 |
| Cardiovascular ds | 12 (11%) | 30 (12%) | 0.860 |
| ICU days | 8.5 ± 9.6 | 12.9 ± 19.6 | 0.004 |
| Hospital days | 19.7 ± 17.3 | 25.5 ± 22.2 | 0.007 |
| SAPS3 | 34.9 ± 6.6 | 35.3 ± 7.6 | 0.672 |
| Initial CVP | 5.3 ± 3.1 | 6.0 ± 3.6 | 0.149 |
| TnI$_1$ | 0.49 ± 1.72 | 0.21 ± 0.81 | 0.134 |
| TnI$_2$ | 2.46 ± 7.78 | 1.33 ± 4.08 | 0.217 |
| TnI$_3$ | 0.82 ± 1.54 | 0.92 ± 5.81 | 0.915 |
| CK-MB | 10.36 ± 23.93 | 6.9 ± 15.6 | 0.161 |
| BNP | 547.4 ± 885.6 | 374 ± 981 | 0.115 |
| cr | 1.4 ± 1.3 | 1.5 ± 1.2 | 0.699 |
| CRP$_{peak}$ | 194.6 ± 96.0 | 221.5 ± 144.7 | 0.072 |
| LVEDD | 44.9 ± 6.4 | 45.8 ± 6.3 | 0.202 |
| LVESD | 29.2 ± 8.1 | 30.5 ± 4.5 | 0.117 |
| LAVI | 19.2 ± 11.7 | 23.0 ± 13.7 | 0.015 |
| LMVI | 92.9 ± 24.9 | 96.7 ± 27.5 | 0.214 |
| E velocity | 70.5 ± 23.4 | 70.4 ± 22.6 | 0.974 |
| A velocity | 85.0 ± 25.6 | 83.3 ± 23.3 | 0.576 |
| DT | 182.0 ± 46.5 | 187.6 ± 46.2 | 0.329 |
| E/E' | 12.1 ± 5.5 | 12.3 ± 5.2 | 0.759 |
| S' | 7.8 ± 2.5 | 7.9 ± 2.4 | 0.641 |
| GS | -16.0 ± 4.9 | -17.3 ± 2.8 | 0.207 |
| MPI | 0.43 ± 0.20 | 0.43 ± 0.20 | 0.910 |
| RVSP | 35.0 ± 10.4 | 33.4 ± 11.2 | 0.199 |

findings or echocardiographic parameters. However, the left atrial volume index (LAVI) was significantly smaller in patients with AEF.

## Multivariate analyses (Table 5)

We performed multivariate analysis in terms of three aspects: 1) presence of hyperdynamic EF, 2) presence of reduced EF, 3) presence of abnormal (both hypo- and hyperdynamic) EF, and 4) in-hospital mortality.

Female sex (OR: 3.316, CI: 1.251–8.789, p = 0.016) and small LV dimension (OR: 0.414, CI: 0.312–0.550, p<0.001) were associated with hyperdynamic LV, and female sex (OR: 1.734, CI: 1.059–2.841, p = 0.029) and small LA size (OR: 0.974, CI: 0.952–0.995, p = 0.018) were

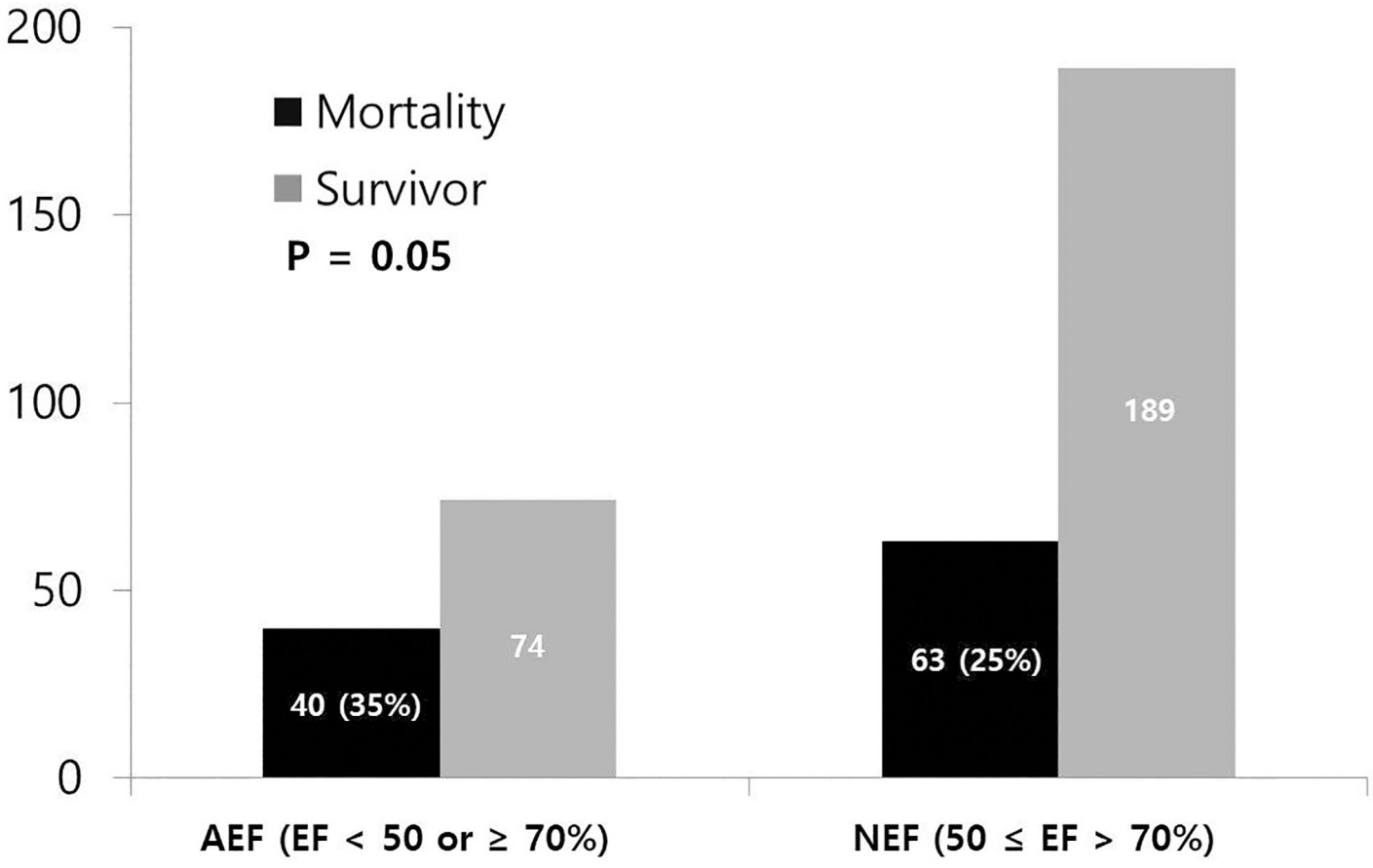

**Fig 3. Comparison of mortality rate according to the EF.** AEF: abnormal ejection fraction, NEF: normal ejection fraction.

associated with the presence of abnormal EF. Tachycardia was associated with the presence of abnormal EF with marginal clinical significance (OR: 1.011, CI: 0.950–1.002, p = 0.064). However, abnormal EF did not show a significant correlation with mortality in multivariate analysis. Tachycardia, pneumonia, presence of malignancy, high SAPS3 score, persistently elevated CRP, and cardiac troponin were associated with mortality in this study. The presence of abnormal EF and its associated factors such as female sex, small LV, or small LA were not associated with mortality in this study; however, tachycardia was associated with mortality.

## Discussion

Reduced LVEF occurred in 36 of 366 (10%) patients and hyperdynamic LVEF occurred in 78 of 366 (21%) patients. Therefore, abnormal (reduced or hyperdynamic) EF occurred in 114 of 336 (31%) patients. Among them, 103 of 366 patients died, resulting in an in-hospital mortality rate of 28%. The patients with abnormal EF showed a higher in-hospital mortality rate than those with normal EF. Abnormal EF was more prevalent in female patients, and slightly increased heart rate, small LA size, and hospital admission days were shorter in patients with abnormal EF. Among them, female sex and small LA size were found to have a significant correlation with the occurrence of abnormal EF in our study.

It has been shown that the early hyperdynamic phase of septic shock is associated with high cardiac output and the late hypodynamic phase is characterized by reduced cardiac output

**Table 5. Multivariate analysis of the associated factors presence of hyperdynamic LV, abnormal LV EF, and mortality.**

| | Odds ratio | 95% CI | p |
|---|---|---|---|
| **Presence of hyperdynamic LV** | | | |
| Female gender | 3.316 | 1.251–8.789 | 0.016 |
| Hospital days | 0.998 | 0.965–1.031 | 0.883 |
| LAVI | 0.974 | 0.928–1.023 | 0.295 |
| LVMI | 0.998 | 0.973–1.023 | 0.855 |
| LVEDD | 0.414 | 0.312–0.550 | <0.001 |
| LVESD' | 0.221 | 0.143–0.342 | <0.001 |
| S' | 1.140 | 0.946–1.373 | 0.168 |
| **Presence of reduced LV EF** | | | |
| Female gender | 1.252 | 0.654–2.397 | 0.498 |
| Mortality | 1.835 | 0.895–2.397 | 0.098 |
| ICU stays | 0.976 | 0.945–1.009 | 0.150 |
| Hospital days | 1.010 | 0.991–2.397 | 1.031 |
| LAVI | 1.014 | 0.992–1.037 | 0.211 |
| **Presence of abnormal LV EF** | | | |
| Female gender | 1.734 | 1.059–2.841 | 0.029 |
| Mortality | 1.460 | 0.829–2.570 | 0.190 |
| HR | 1.011 | 0.999–1.022 | 0.064 |
| ICU stays | 0.978 | 0.950–1.006 | 0.117 |
| Hospital days | 1.000 | 0.983–1.016 | 0.963 |
| LAVI | 0.974 | 0.952–0.995 | 0.018 |
| **Mortality** | | | |
| HR | 1.017 | 1.005–1.029 | 0.004 |
| Pneumonia | 3.184 | 1.817–5.580 | <0.001 |
| Malignancy | 2.949 | 1.534–5.669 | 0.001 |
| SAPS3 | 1.060 | 1.015–1.106 | 0.008 |
| Final CRP | 1.019 | 1.010–1.028 | <0.001 |
| Final TnI | 1.657 | 1.045–2.630 | 0.032 |

[19–23]. In this study, EF was measured in patients who underwent echocardiography within 48 hours of hospitalization for sepsis, and patients were classified into three groups according to the result. Thirty-six (10%) patients showed reduced EF ($< 50\%$) and 78 (21%) patients showed increased EF ($\geq 70\%$). Therefore, according to the results of this study, cardiac function may decrease or increase from the time of admission. Of course, one disadvantage of this study was that the patients were not admitted to hospital at the same stage of sepsis, so we could not determine whether they visited our hospital in the early stages of the illness or whether they had deteriorated during treatment at the other hospital. However, whether the patient was staying in a nursing home or in their home did not have a significant effect on EF reduction or exertion. Most previous studies have investigated patients with sepsis and reduced cardiac function (sepsis-related CMP) or compared patients with hyperdynamic LV (increased EF) and normal EF. The findings of this study are meaningful in that the mortality rate was higher in patients with abnormal EF (reduced or increased) as measured by echocardiographic confirmation of cardiac function within 48 hours of admission. In addition, the exclusion of patients with underlying cardiac disease through questionnaires or chart reviews is an important strength in assessing cardiac dysfunction due to sepsis.

Cardiac specific biomarkers such as Troponin I have been studied in septic patients and elevated Troponin I has been reported as associated with higher risk of mortality [24, 25]. We also investigated the relationship between abnormal EF and cardiac troponin, but there was no significant difference between EF and cardiac troponin elevation. One-way ANOVA showed that cardiac troponin increased with decreasing EF and at its lowest when EF increased. However, there was no significant difference in cardiac troponin among the three groups according to the Kruskal-Wallis test. It has been well known that acute bacterial infections are associated with an increased risk of myocardial infarction [26], and further cardiac or coronary evaluation for statins and antiplatelet therapy may be needed in survived patients with persistently elevated cardiac troponin. BNP levels were similar (less than 400 pg/mL) in patients with normal EF and elevated EF, but the BNP level was elevated to nearly 1,000 pg/mL in patients with reduced EF. The BNP level helps to rule out or diagnose heart failure (HF) in patients with dyspnea. A BNP level less than 100 pg/mL means less chance of HF, and a BNP level greater than 400 pg/mL increases the likelihood of HF [27, 28]. If the BNP level is between 100 and 400 pg/mL, other causes besides HF should be considered under clinical judgment, and BNP may be elevated in the case of sepsis [27–29]. Natriuretic peptides are protein molecules that are secreted by the ventricular musculature in response to volume or pressure overload [30]; therefore, this is considered to be a secondary response to lowered cardiac function in patients with SIC. In addition, unlike cardiac troponin, BNP levels did not correlate with mortality in our study.

The etiology of sepsis may affect the type of cardiac dysfunction. To explain it further, sepsis due to underlying lung pathology can result in elevation of RV afterload. High RV afterload leads to less pulmonary blood flow and reduces the possibility of left ventricular (LV) diastolic failure if LV function is normal prior to sepsis [19, 31–34]. In this study, neither the etiology of sepsis (infection site), nor the presence of underlying lung disease, affected the occurrence of AEF. However, patients with pneumonia had higher right ventricular systolic pressure (RVSP) measured on echocardiography than those with other causes of sepsis (35.9 ± 12.4 vs. 33.0 ± 9.0, p = 0.012). In terms of mortality, patients with sepsis due to pneumonia had a higher mortality rate than those with other causes (OR: 3.184, 95% CI: 1.817–1.029, p<0.001).

Central venous pressure (CVP) has been used to assess hemodynamic status, particularly in the ICU [35]. A normal CVP is between 8 to 12 mmHg, this value can be altered by volume status or venous compliance [36, 37]. EF can also be affected by preload [38], so we compared CVP between the three groups. However, there was no significant difference in terms of EF. There is a question as to whether the CVP itself accurately reflects the preload [35, 36], and there is no assurance that it has been accurately measured in all patients in this study. Therefore, it cannot be asserted that CVP is not a relevant factor for determining EF in patients with sepsis, and it should be clarified by further studies.

## Limitations

This study had several limitations. Firstly, this was a single center study with a relatively small study population. Secondly, in this study, echocardiography was performed 48 hours after admission, so patients with late onset AEF were excluded. Thirdly, serial echocardiographic follow-up was not performed, except in patients with reduced EF. Therefore, we could not analyze the effects of changes in EF and clinical implications in patients with sepsis over time. Among the patients who showed reduced EF, echocardiography was followed up for 7 to 10 days if possible, and most of the patients recovered to normal cardiac function. However, restoration of cardiac function did not affect the mortality rate. Finally, a relatively small study population was one of the major limitations. Therefore, serial echocardiography follow-up

should be performed in patients who underwent echocardiography within 48 hours of admission. Further studies should focus on the effects of cardiac function over time and clinical outcomes.

In conclusion, our findings highlight that female sex and small cardiac size are associated with abnormal EF, and therefore, death. Female patients and patients with small LA should be monitored closely when they present with sepsis. Our findings can assist clinicians to provide better patient care.

## Supporting information

**S1 Checklist.**
(DOT)

**S1 Dataset.**
(SAV)

## Author Contributions

**Conceptualization:** Min-Kyung Kang.

**Data curation:** Min-Kyung Kang, Yu Bin Seo, Jaehuk Choi, Seon Yong Choi, Seonghoon Choi, Jung Rae Cho, Namho Lee.

**Formal analysis:** Min-Kyung Kang.

**Investigation:** Min-Kyung Kang.

**Methodology:** Min-Kyung Kang.

**Software:** Min-Kyung Kang.

**Supervision:** Min-Kyung Kang.

**Writing – original draft:** Dong Geum Shin.

**Writing – review & editing:** Min-Kyung Kang.

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
