## [Decision Letter · Decision Letter 0]

27 Jan 2020

PONE-D-19-32205

Factors Associated with Abnormal Left Ventricular Ejection Fraction (Decreased or Increased) in Patients with Sepsis in the Intensive Care Unit

PLOS ONE

Dear Dr Kang,

Thank you for submitting your manuscript to PLOS ONE. After careful consideration, we feel that it has merit but does not fully meet PLOS ONE’s publication criteria as it currently stands. Therefore, we invite you to submit a revised version of the manuscript that addresses the points raised during the review process.

We would appreciate receiving your revised manuscript by Mar 12 2020 11:59PM. To enhance the reproducibility of your results, we recommend that if applicable you deposit your laboratory protocols in protocols.io, where a protocol can be assigned its own identifier (DOI) such that it can be cited independently in the future. For instructions see: http://journals.plos.org/plosone/s/submission-guidelines#loc-laboratory-protocols

We look forward to receiving your revised manuscript.

Kind regards,

Chiara Lazzeri

Academic Editor

PLOS ONE

Journal Requirements:

'No. The funders had no role in study design, data collection and analysis, decision to publish, or preparation of the manuscript.'

Please provide an amended Funding Statement that declares *all* the funding or sources of support received during this specific study (whether external or internal to your organization) as detailed online in our guide for authors at http://journals.plos.org/plosone/s/submit-nowPlease state what role the funders took in the study.  If any authors received a salary from any of your funders, please state which authors and which funder. If the funders had no role, please state: "The funders had no role in study design, data collection and analysis, decision to publish, or preparation of the manuscript."

4. We note you have included a table to which you do not refer in the text of your manuscript. Please ensure that you refer to Table 5 in your text; if accepted, production will need this reference to link the reader to the Table.

5. Please include captions for your Supporting Information files at the end of your manuscript, and update any in-text citations to match accordingly. Please see our Supporting Information guidelines for more information: http://journals.plos.org/plosone/s/supporting-information

Reviewers' comments:

Reviewer's Responses to Questions

**Comments to the Author**

1. Is the manuscript technically sound, and do the data support the conclusions?

Reviewer #1: Partly

Reviewer #2: Partly

2. Has the statistical analysis been performed appropriately and rigorously? 

Reviewer #1: Yes

Reviewer #2: No

3. Have the authors made all data underlying the findings in their manuscript fully available?

Reviewer #1: Yes

Reviewer #2: No

4. Is the manuscript presented in an intelligible fashion and written in standard English?

Reviewer #1: No

Reviewer #2: Yes

5. Review Comments to the Author

Reviewer #1: Hypothesis- Study was designed to look for risk factors associated with septic cardiomyopathy

In results - Grouping of hyperdynamic and abnormal EF group is not clear, as abnormal EF group includes both hypo and hyperdynamic EF

So interpretation of results is difficult

Need to analyse low EF and hyperdynamic EF separately

Classification of comorbidities was done based on what ?

Supplementary material is not in english, so not able to evaluate

Needs plagiarism check

Reviewer #2: I thank the authors and the editor for allowing me to review this manuscript. In “Factors Associated with Abnormal Left Ventricular Ejection Fraction (Decreased or Increased) in Patients with Sepsis in the Intensive Care Unit,” Shin and colleagues assessed septic patients and categorized them by ejection fraction. They found that hearts with abnormal ejection fractions were associated with increased mortality. Additionally, they found female sex and decreased chamber size was associated with increased ejection fraction. Although not a criteria for PLOS One publication, this study has substantial interest to clinicians and researchers. I will address the PLOS one criteria before moving on to major and minor comments:

To be accepted for publication in PLOS ONE, research articles must satisfy the following criteria:

1. The study presents the results of original research.

Yes. While similar studies exist, this particular manuscript identifies a different patient population, with different results than those published previously.

2. Results reported have not been published elsewhere.

Yes. A brief pubmed and google search reveals no suggestion of duplicate publication.

3. Experiments, statistics, and other analyses are performed to a high technical standard and are described in sufficient detail.

No. There are multiple methodological errors. However, these can be remedies.

4. Conclusions are presented in an appropriate fashion and are supported by the data.

Questionable. I suspect the data will continue to support conclusions after correcting the methods.

5. The article is presented in an intelligible fashion and is written in standard English.

Yes. The article is intelligible and uses standard English

6. The research meets all applicable standards for the ethics of experimentation and research integrity.

Yes. The study meets face validity for ethical research, and authors attest it is in accordance with ethical standards, and approved by an IRB.

7. The article adheres to appropriate reporting guidelines and community standards for data availability.

This is unclear. The authors attest that data are publicly available, but the description of where the Data may be found is listed as “no”

Major Comments:

1. Patient selection is unclear in the methods. My read suggests that all patients who were in the ICU with sepsis received an echocardiogram and a cardiology consultation. This seems unusual, as even hospitals with high echocardiography utilization don’t perform echocardiography on 100% of septic patients. I would expect a number of patients admitted with sepsis, and a subset of those patients received a cardiac consultation and echocardiogram. Please clarify.

2. The authors do not specify how sepsis was determined. Sepsis 3 criteria were disseminated in early 2016. Were these patients categorized according to Sepsis 2, Sepsis 3, a combination? Were they categorized by some screening criteria, or was assignment of sepsis based on a clinical gestalt, medical documentation, or billing? Please clarify.

1. It is unclear how some of the cardiac patients were excluded. Stress-induced cardiomyopathy can overlap with septic cardiomyopathy. There is disagreement whether septic cardiomyopathy and stress induced cardiomyopathy represent distinct phenotypes. Additionally, the exclusion of antecedent LV dysfunction or restrictive cardiomyopathy likely results in a selection bias, as patients who have had prior echocardiograms are different than those who have not. There is no such thing as ischemic cardiomyopathy. The WHO definition for cardiomyopathy excludes ischemia. The preferred term is ischemic heart disease. Ordinarily, this would just be a pedantic comment, but the inclusion of this term makes me wonder about whether hypertrophic cardiomyopathy is being being conflated with hypertrophic heart—terms clinicians often throw about casually. To be clear, hypertrophic cardiomyopathy is the autosomal dominant disease of excess muscle growth, despite that the term is often used erroneously to describe hypertrophy of hypertension or valvular disease. Regardless, please include a strobe flow diagram that details number of patients included and excluded.

3. The study is retrospective, but the description of the echoes sounds more like a detailed prospectively performed protocol. If the echoes were clinically obtained echoes, please simply state that the echoes were performed clinically in adherence with guidelines by appropriately certified people. Additionally, please clarify that the measurements were clinically obtained by the sonographer or physician, unless the investigators performed additional measurements as part of the research investigation.

4. As data are non-normally distributed, it is customary to report median and interquartile ranges instead of mean and standard deviation for those data.

5. The authors perform multiple comparisons without an a priori hypothesis. I see scores of comparisons. Please correct for multiple hypothesis testing using Bonferroni or some other similar correction.

6. 48 hours after sepsis is a long time. The authors do not mention accounting for censoring by death within the first 48 hours. Additionally, censoring by death prevents most echocardiographic follow-up. The STROBE diagram might help demonstrate how many patients might have died before having an opportunity to receive echocardiogram.

7. I would like to see references and comparison to other similar studies in the field. Prior investigations have reported lower incidence of hyperdynamic hypovolemia, and higher incidence of hypocontractility. I suspect this finding is due both to selection criteria, which excluded all patients with antecedent EF<50%, and practice variation where the study population may perhaps receive less volume expansion than those other hospitals. Antecedent fluid receipt may be a more direct predictor of hyperdynamic hypovolemia. Additionally, vasopressors and ventilator receipt could be obvious confounders.

8. Ventricular function can be affected by so many parameters. Other studies in this field had accounted for vasopressor dose receipt of mechanical ventilation at time of echo, and antecedent fluid receipt. This study should adjust for these confounders.

Minor Comments:

1. Page 8. “...were measured at end-diastole according to the standards established by the American Society of Echocardiography2.” Please omit the 2. Additionally, the guideline cited by Porter et al. is out of date and has been replaced by a more recent publication.

2. Table 1-3. Group 3 is misspelled.

6. PLOS authors have the option to publish the peer review history of their article (what does this mean?). If published, this will include your full peer review and any attached files.

Reviewer #1: Yes: Dr Amarja Ashok Havaldar

Reviewer #2: No

---

## [Author Response · Author response to Decision Letter 0]

30 Jan 2020

I added a file titled as "response to reviewer's"

---

## [Editor Report · Decision Letter 1]

11 Feb 2020

Factors Associated with Abnormal Left Ventricular Ejection Fraction (Decreased or Increased) in Patients with Sepsis in the Intensive Care Unit

PONE-D-19-32205R1

Dear Dr. Kang,

We are pleased to inform you that your manuscript has been judged scientifically suitable for publication and will be formally accepted for publication once it complies with all outstanding technical requirements.

With kind regards,

Chiara Lazzeri

Academic Editor

PLOS ONE
---

## [Editor Report · Acceptance letter]

26 Feb 2020

PONE-D-19-32205R1 

Factors Associated with Abnormal Left Ventricular Ejection Fraction (Decreased or Increased) in Patients with Sepsis in the Intensive Care Unit 

Dear Dr. Kang:

I am pleased to inform you that your manuscript has been deemed suitable for publication in PLOS ONE. Congratulations! Your manuscript is now with our production department. 

With kind regards,

on behalf of

Dr. Chiara Lazzeri 

Academic Editor

PLOS ONE